# Trend of genital ulcers and discharge and associated factors among survey respondents in Tanzania, 2004–2022: Analysis of demographic health surveys

**Zuhura Mbwana Ally**[1]*, **Lynn Moshi**[2], **Rahma Musoke**[3], **Mariam Salim Mbwana**[4], **Hafidha Mhando Bakari**[5], **Swalehe Mustafa Salim**[6], **Leticia Francis Karia**[7], **Maximillian Francis Karia**[8], **Alpha Johnson Kapola**[9], **Glenda Marie Manayon**[10], **Haji Mbwana Ally**[11], **Hassan Fredrick Fussi**[12], **Habib Omari Ramadhani**[13]

**1** District Hospital, Tanga, Tanzania, **2** Aga Khan Hospital, Dar es salaam, Tanzania, **3** Water mission, Tanzania, **4** Primary Health Care Institute, Iringa, Tanzania, **5** University of Dar es salaam, Dar es salaam, Tanzania, **6** Canada Youth Group, Dar es salaam, Tanzania, **7** Appalachian State University, Boone, North Carolina, United States of America, **8** The Greenfield School at Wilson, North Carolina, United States of America, **9** EMET Healthcare, Dar es salaam, Tanzania, **10** San Lazaro Hospital, Manila, Philippines, **11** Kilimanjaro Christian Medical Center, Tanzania, **12** District Hospital, Dar es salaam, **13** Institute of Human Virology, University of Maryland School of Medicine, Baltimore, Maryland, United States of America

* zuhuraally86@gmail.com

## Abstract

### Introduction

Sexually transmitted infections (STIs) continue to pose a significant public health challenge worldwide, with over 1 million new cases reported daily. STI's are known to negatively impact sexual and reproductive health, increases risk of HIV transmission and acquisitions, as well as impacting mental health, personal wellbeing, and relationships. Information on the trend of STIs using nationally representative data in Tanzania is lacking. We evaluated trends in the prevalence of genital ulcers and discharge using the 2004, 2010 and 2022 Tanzania Demographic Health Survey (TDHS) data.

### Materials and Methods

The TDHS are nationally representative, cross-sectional household surveys that used a two-stage cluster-based sampling design, selecting enumeration areas followed by households. Participants aged 15–49 years who responded to questions on whether they have ever had genital ulcers and/or discharge in the last 12 months prior to the survey were included in this analysis. Survey collected sociodemographic characteristics, HIV testing, STIs symptoms, number of sex partners and recency of sexual activity. We quantified weighted prevalences, and hierarchical mixed effects multilevel logistic regression models that accounted for survey weights, stratification and clustering were used to compute adjusted odds ratios (aOR) and 95% confidence intervals (CI) for factors associated with genital ulcers/discharge.

**Data availability statement:** Third party data was obtained for this study from The DHS Program (https://dhsprogram.com/). Data may be requested from The DHS Program after creating an account and submitting a concept note. More access information can be found on The DHS Program website (https://dhspro-gram.com/data/Access-Instructions.cfm).The data set is openly available upon permission from the MEASURE DHS website (https://www.dhsprogram.com/data/available-datasets.cfm). The authors confirm that interested researchers would be able to access these data in the same manner as the authors. The authors also con-firm that they had no special access privileges that others would not have.

**Funding:** The author(s) received no specific funding for this work.

**Competing interests:** The authors have declared that no competing interests exist.

## Results

Among 46,481 participants with a median age of 27 (interquartile range: 20–36) years, the overall prevalence of genital ulcers/discharge was 6.7%; (4.3% in 2004, 5.0% in 2010 and 9.3% in 2022). Generally, there was an increased trend in genital ulcers/discharge with the increase being higher between 2010 and 2022 compared to 2004 and 2010. In the full adjusted model, females (aOR=1.68: 95%CI 1.48–1.90) had higher odds of genital ulcers/discharge. Divorced/separated/widowed, young individuals, being sexually active 4 weeks prior to the survey and higher number of lifetime sex partners were all associated with higher odds of genital ulcers/discharge.

## Conclusions

There was an increase in the trend of prevalence of genital ulcers/discharge in Tanzania with women and divorced/separated or widowed individuals being disproportionally affected. Identifying reasons for the increase in these STI related symptoms is paramount to strategize and address challenges. Policymakers should prioritize funding for STI diagnostic tools and community education programs. Additionally, healthcare facilities should incorporate routine STI screenings, particularly for high-risk groups like women and those with multiple sexual partners, to curb the rising prevalence.

## Introduction

Sexually transmitted infections (STIs) are known to negatively impact people in many ways. In reproductive health, STIs can lead to abortions, still birth, infertility, ectopic pregnancy and pelvic inflammatory diseases [1–3]. STIs increases risk of HIV transmission and acquisitions [4]. Moreover, STIs increase the risk of neurological diseases [5], as well as impacting mental health, personal wellbeing and relationships [6]. Combating STIs may mitigate these negative consequences.

In 2020, the World Health Organization estimated 374 million new infections involving gonorrhea, chlamydia, syphilis and trichomoniasis. The estimated rate of acquiring these curable infections among people aged 15–49 years was > 1 million per day [7].

Prospectively collected data from eight health care facilities in Tanzania between 2014 and 2016 showed that the prevalence of gonorrhea, chlamydia and mycoplasma among patients presented with genital symptoms were 25.7%, 12.9 and 4.7%, respectively [8]. The 2004 Tanzania demographic and health survey also indicated that, 2% of men and 3% of women self-reported having had discharge or genital sores and the prevalences of these STI related symptoms for men and women in 2010 survey were 6% and 7%, respectively [9,10]. These data indicate a high burden of STIs globally and regionally. Like in many resources limited countries, Tanzania uses syndromic management of STIs, however, syndromic management has limitation in the detection and management of asymptomatic cases[8]. Systematic review showed that the sensitivity and specificity of genital ulcer disease in detecting syphilis were 52.8% and 72.1%; herpes were 43.5% and 88.0%; and chancroid were 71.9% and 53.1% respectively [11].

Although most of these STIs are curable, several individual and health care system challenges exist. Some individual challenges include lack of awareness of STIs and negative impact of these infections, stigmatization, and attitudes towards behavior change [12–14]. Health care system challenges include lack of accurate diagnostic tools, technical capacity of health care workers and stigmatization [15,16]. These barriers deter efforts to control STIs.

Prior data have shown several factors are associated with increased risk of STI acquisition and transmission. These include concurrent partnerships [17], high number of lifetime partners [18], young age at sexual debut, greater age differences between partners [19], alcohol and other substance use [20], inconsistent condom use and partner violence [21]. Exploring additional factors that may be associated with increased risk of STI related symptoms is critical to understand targeted interventions.

Despite the public health significance of STIs, most studies in Africa have only reported prevalence of these infections [8,22,23]. Although these data are important, they do not provide the magnitude of STIs related symptoms overtime. Data on the trend of symptomatic STI prevalence is important for STI control. Due to the lack of data on the trend of symptoms of STIs in Tanzania, we aimed to explore trend in the prevalence of STI related symptoms among people aged 15–49 years using the 2004, 2010 and 2022 Tanzania Demographic and Health Survey (TDHS) data.

## Materials & methods

### Survey design

The TDHS are nationally representative, cross-sectional household (HH) surveys conducted in 2004, 2010 and 2022. TDHS used a two-stage cluster-based sampling design, selecting enumeration areas followed by HH as previously described [9,10,24]. Among other things, sociodemographic characteristics, HIV testing, symptoms of STIs, sexual partners and recency of sexual activities data were collected from these surveys.

### Inclusion and exclusion criteria

All survey participants aged 15–49 years who participated in the TDHS and responded to questions about whether they ever had genital ulcers and/or discharge in the 12 months prior to survey were included in this analysis. Survey participants with missing information about STI symptoms in the 12 months prior to survey were excluded. The 2015 TDHS dataset did not have variables related to abnormal discharge and genital ulcers and therefore not included in the analysis.

### Variables and definitions

A person was considered to ever had genital ulcers/discharge if he/she reported the presence of genital ulcers and/or discharge 12 months prior to survey. We dichotomized age into two groups: 15–24 years (youths) and 25–49 years, education was categorized as no education, primary education or higher. Marital status was categorized as never married, married/ living together and divorced/separated/widowed, and residence was categorized as urban or rural. Wealth index was used to categorize wealth index scores into quintiles (lowest, second, middle, and highest) as previously described [9]. Based on the distribution, number of lifetime sex partners was categorized as 1–2, 3–4 or >4 partners. Sexual activity was categorized as never had sex, sexually active in the last 4 weeks prior to survey and not sexually active in the last 4 weeks prior to survey. STI awareness was also measured as aware or not aware, and a wife's ability to request that her husband use a condom if he had an STI was also measured as previously described [10]. Age of sexual debut was categorized as <15 years, 15–19 years and ≥20 years. Based on the distribution, the number of alcoholic drinks per day was categorized into three groups (0–1, 2–3 and ≥4). Survey year was also included (2004, 2010 and 2022). Finally, HIV testing information was categorized as ever tested for HIV and received test results vs never tested.

## Statistical analysis

We verified data sets for completeness and consistency. These datasets were then cleaned, coded and appended for pooled analysis [9,10,24]. Proportions of participant covariate characteristics were stratified by the survey year and weighted percentages were tabulated. DHS used variables related to participant selection probabilities at each stage of sampling (cluster, household, individual), and response rates at the household and individual level to calculate weights. These were then adjusted for non-response to create the final survey weights. Realizing hierarchical structure of the data, i.e., survey respondents are nested within survey year, mixed effects multilevel logistic regression models were used to analyze data. In this analysis, individual characteristics were considered as level-1 and survey year as level-2 variables. The mixed effect models consisted of fixed and random effects. The fixed effects represent factors associated with genital ulcers/discharge quantified by adjusted odds ratio (aOR) and 95% confidence intervals (CI). Random effects represents variation in the probability of having genital ulcers/discharge. The random effects was represented by intraclass correlation coefficient (ICC) which indicates how much of the total variation in the probability of having genital ulcers/discharge is accounted for by the survey year. We computed ICC using the formular $ICC = [Q_{survey\_yr}/(Q_{survey\_yr} + Q_{error})]$, with $Q_{survey\_yr}$ and $Q_{error}$ indicating variance estimates of survey year and error respectively. In modeling a binary response in a hierarchical structure, it is assumed that no error term for level-1 variables and a variance of 3.29 was used in place of $Q_{error}$ for the computation or ICC as previously described [25]. For the multivariable analysis, three hierarchical models were fit. Model one included intercept only; model 2 included level-1 variables (individual characteristics). In model 3 we added level -2 variable (survey year) to model 2. −2Log likelihood (−2LL) and likelihood ratio test were used to assess the fitness of hierarchical models. Logistic regression models accounted for survey weights, stratification and clustering in the sample design. All variables evaluated on the bivariate analysis, except number of sex partners and alcohol drinks per day, were then included in the final multivariable model. Due to the correlation between lifetime sex partners and number of sex partners, we chose to include lifetime sex partners in the multivariable models because of its statistically significant association in the bivariate model, and it had fewer missing data. We excluded the number of alcoholic drinks per day from the multivariable analysis due to the limited number of responses, as this variable was only collected in the 2022 survey. Complete case analysis was performed in the regression model. A p-value of < 0.05 was used to declare statistical significance. All analyses were conducted in SAS 9.4 (Cary, NC).

## Ethical considerations

Secondary non-identifiable survey data that is publicly available to registered users from the online data repositories was used for this study. These demographic health surveys received approval from the Tanzania National Institute of Medical Research and the Institutional Review Board of ICF International. All adult respondents gave informed consent and, for minors, written informed consent was first obtained from the parent or guardian followed by written assent by the participants. Authors of this manuscript submitted a proposal to the DHS Program/ICF International and received permission to download and use the data for this study. The DHS Program authorized data access and the data were used solely for the purpose of the current study. DHS data is publicly available at https://dhsprogram.com/ upon request.

## Results

### Characteristics of the study participants

Of 46,647 participants aged 15–49 years, 166 (0.4%) were not sure if they had ever had genital ulcers and/or discharge 12 months prior to the survey and, therefore, were excluded from this

analysis. The final analysis included 46,481 participants whose median age was 27 (interquartile range (IQR): 20–36) years. Overall, a total of 6.7% survey participants reported ever having genital ulcers/discharge (4.3% in 2004, 5.0% in 2010 and 9.3% in 2022). Sixty percent of survey participants were 25 years or older, 76.6% were female and 82.8% had primary school education or higher. Nearly 69% were rural residents, 60.6% were married or living together and 83.4% were on the second wealth quantile or higher. Fifty-two percent of participants had ever tested for HIV and received test results, 53.8% had one-to-two lifetime sexual partners and 55.8% were sexually active within the four weeks prior to the survey (Table 1).

## Changes in the prevalence of genital ulcers/discharge in relation to participant characteristics

Table 2 summarizes the changes in the prevalence of genital ulcers/discharge in relation to participant characteristics. The assessment of changes in the prevalence of genital ulcers/ discharge was categorized into two phases: between the 2004 and 2010 surveys (phase 1), and between the 2010 and 2022 surveys (phase 2). Overall, there was an increase in the prevalence of genital ulcers/discharge from 2004 to 2022 by 5.0%. Generally, a consistent increase in prevalence of genital ulcers/discharge in both phases was exhibited across almost all participant characteristics. The increase in the prevalence of genital ulcers/discharge was higher in phase II compared to phase I. For example, among participants aged 25–49 years, the increase was 0.6% in phase I compared to 5.1% in phase II. Females had a 0.9% increase in phase I and 4.8% increase in phase II. Divorced/separated/widowed participants had a 1.4% increase in genital ulcers/discharge prevalence in phase I compared to 7.5% in phase II. Being sexually active in the four weeks prior to the survey was associated with 1.1% increase in the prevalence of genital ulcers/discharge in phase I and 5.1% in phase II.

## Factors associated with the prevalence of symptomatic STIs: bivariate analysis

Compared to youths, participants who were 25 years or older had higher odds of having genital ulcers/discharge (OR = 1.65; 95% CI 1.48–1.63) (Table 3). Other participant characteristics that were associated with higher odds of genital ulcers/discharge included gender, female vs male (OR = 1.16: 95%CI 1.02–1.31); marital status, divorced/separated/widowed vs never married (OR=3.12: 95%CI 2.62–3.71) and married/living together vs single (OR=2.16: 95%CI 1.87–2.50); number of lifetime sex partners, 3–4 vs 1–2, (OR=1.67: 95%CI 1.46–1.91) and >4 vs 1–2 (OR=2.04: 95%CI 1.77–2.35); ever tested for HIV and received test results compared to never tested (OR=2.27: 95%CI 2.02–2.56); and recent sexual activity, active within 4 weeks vs not active within 4 weeks prior to the survey (OR=1.28: 95%CI 1.16–1.42). Furthermore, compared to survey respondents in 2004, those in 2022 had higher odds of genital ulcers/discharge (OR=2.30: 95%CI 2.00–2.64).

## Factors associated with prevalence of genital ulcers/discharge: multivariable analysis

-2log likelihood and likelihood ratio test showed that fully adjusted model (model 3) was better than other two models and results presented below are from fully adjusted model. Results of the fully adjusted model showed that age, gender, marital status, number of lifetime sex partners, recent sexual activity, partner's justification in requesting their husband to use a condom if he has had an STI and survey year consistently exhibited significant associations with having had genital ulcers/discharge. For example, females had 68% higher odds of genital

**Table 1. Characteristic of participants aged 15-49 years who responded to questions on sexually transmitted infections from demographic surveys in 2004, 2010 and 2022 in Tanzania.**

| Characteristics | Overall | 2004 | 2010 | 2022 |
|---|---|---|---|---|
| | (n = 46,481) | (n = 12,917) | (n = 12,654) | (n = 20,910) |
| | n (%) | n (%) | n (%) | n (%) |
| Age | | | | |
| 15–24 | 18,788 (40.1) | 5,372 (41.5) | 5,180 (40.6) | 8,236 (39.0) |
| 25–49 | 27,693 (59.9) | 7,545 (58.5) | 7,474 (59.4) | 12,674 (61.0) |
| Sex | | | | |
| Male | 10,862 (23.4) | 2,607 (20.2) | 2,523 (19.9) | 5,732 (17.4) |
| Female | 35,619 (76.6) | 10,310 (79.8) | 10,131 (80.1) | 15,178 (72.6) |
| Education | | | | |
| No education | 7,937 (17.2) | 2,847 (21.7) | 2,138 (17.2) | 2,952 (14.3) |
| ≥ Primary education | 38,544 (82.8) | 10,070 (78.3) | 10,516 (82.8) | 17,958 (85.7) |
| Marital status | | | | |
| Never married | 14,262 (29.2) | 3,636 (26.7) | 3,838 (28.3) | 6,788 (31.3) |
| Married/living together | 27,657 (60.6) | 8,141 (64.5) | 7,560 (61.0) | 11,956 (57.9) |
| Divorced/separated/widowed | 4,562 (10.2) | 1,140 (8.8) | 1,256 (10.7) | 2,166 (10.8) |
| Residence | | | | |
| Urban | 13,598 (31.3) | 3,102 (28.2) | 3,215 (28.3) | 7,281 (35.1) |
| Rural | 32,883 (68.7) | 9,815 (71.8) | 9,439 (71.7) | 13,629 (64.9) |
| Wealth quantile | | | | |
| Lowest | 7,306 (16.6) | 2,240 (17.9) | 1,992 (16.4) | 3,074 (15.9) |
| Second | 8,222 (18.1) | 2,385 (18.9) | 2,335 (18.9) | 3,502 (17.2) |
| Middle | 9,041 (19.3) | 2,347 (19.0) | 2,388 (19.6) | 4,306 (19.4) |
| Forth | 10,514 (21.3) | 2,938 (19.5) | 2,879 (21.2) | 4,697 (22.4) |
| Highest | 11,398 (24.7) | 3,007 (24.7) | 3,060 (23.9) | 5,331 (25.1) |
| Tested for HIV and received results[*] | | | | |
| No | 22,704 (48.0) | 11,324 (87.7) | 6,059 (46.7) | 5,321 (24.6) |
| Yes | 23,374 (52.0) | 1,411 (12.3) | 6,374 (53.3) | 15,589 (75.4) |
| Number of lifetime sexual partners[*] | | | | |
| 1–2 | 15,647 (53.8) | N/A | 6,564 (60.8) | 9,083 (49.5) |
| 3–4 | 6,301 (24.5) | N/A | 2,284 (24.0) | 4,017 (24.8) |
| >4 | 5,399 (21.7) | N/A | 1,394 (15.2) | 4,005 (25.6) |
| Number of partners/wives[*] | | | | |
| 0 | 17,360 (63.7) | 5,132 (63.6) | 4805 (64.2) | 7,423 (63.4) |
| 1 | 8,648 (30.5) | 2,458 (29.6) | 2279 (29.3) | 3,911 (31.8) |
| >1 | 1,635 (5.8) | 541 (6.8) | 472 (6.5) | 622 (4.8) |
| Recent sexual activity | | | | |
| Never had sex | 8,412 (14.8) | 2,210 (13.7) | 2,397 (14.9) | 3,805 (15.5) |
| Active in the last 4 weeks | 25,168 (55.8) | 6,949 (55.9) | 6,885 (56.2) | 11,334 (55.6) |
| Not active in the last 4 weeks | 12,885 (29.4) | 3,751 (30.5) | 3,363 (28.9) | 5,771 (29.0) |
| STI | | | | |
| No | 43,811 (93.3) | 12,448 (95.7) | 12,146 (95.0) | 19,217 (90.7) |
| Yes | 2,670 (6.7) | 469 (4.3) | 508 (5.0) | 1,693 (9.3) |
| Ever heard of STI | | | | |
| No | 4,945 (9.9) | 79 (0.7) | 38 (0.4) | 4,828 (21.4) |
| Yes | 38,929 (84.5) | 10,231 (79.1) | 12,636 (99.6) | 16,082 (78.6) |
| Unknown | 2,607 (5.6) | 2,607 (20.2) | 0 (0.0) | 0 (0.0) |

*(Continued)*

**Table 1.** (Continued)

| Characteristics | Overall | 2004 | 2010 | 2022 |
|---|---|---|---|---|
| | (n = 46,481) | (n = 12,917) | (n = 12,654) | (n = 20,910) |
| | n (%) | n (%) | n (%) | n (%) |
| Age at first sex | | | | |
| <15 | 13,651 (27.0) | 3,830 (26.4) | 3,823 (27.4) | 5,998 (27.3) |
| 15–19 | 25,469 (58.8) | 7,081 (59.6) | 6,725 (57.2) | 11,663 (59.3) |
| ≥20 | 7,356 (14.1) | 2,002 (14.0) | 2,105 (14.6) | 3,249 (13.4) |
| Number of alcohol drinks per day* | | | | |
| 0–1 | 429 (21.7) | N/A | N/A | 429 (21.7) |
| 2–3 | 895 (45.3) | N/A | N/A | 895 (45.3) |
| ≥4 | 604 (33.0) | N/A | N/A | 604 (33.0) |
| Wife request husband to use condom if he had STI | | | | |
| No | 10,254 (20.7) | 1,826 (13.9) | 1,958 (14.0) | 6,470 (28.9) |
| Yes | 30,692 (68.5) | 7,624 (59.7) | 10,009 (81.6) | 13,059 (66.0) |
| Unknown | 5,535 (10.8) | 3,467 (26.4) | 687 (4.4) | 1,381 (5.1) |

Abbreviations:

*indicates number of people assessed were less than 46,481; n indicates number of people; HIV indicates human immune deficiency virus; N/A indicates not applicable

ulcers/discharge compared to males (aOR=1.68: 95%CI 1.48–1.90) (Table 4). Divorced/separated/widowed participants and those who were sexually active within the four weeks prior to the survey had 17% and 28% higher odds of genital ulcers/discharge compared to single and those who were not sexually active within the four weeks prior to the survey, respectively. Compared to participants who had 1–2 lifetime sex partners, those who had 2–4 (aOR=1.61: 95%CI 1.44–1.80) and >4 (aOR=2.20: 95%CI 1.94–2.50) had higher odds of genital ulcers/discharge. Similarly, survey respondents in 2022 had higher odds of genital ulcers/discharge compared to those in 2004 (aOR=2.27: 95%CI 2.00–2.52).

Participants whose age of sexual debut was 20 years of age or older had lower odds of genital ulcers/discharge compared to those whose age of sexual debut was less than 15 years (aOR=0.85: 95%CI 0.74–0.99).

## Random effects results

Across all hierarchical models, four to five percent of variability in the probability of having genital ulcers/discharge was accounted for by survey year, leaving behind 95–96 percent of the variability in the probability of having genital ulcers/discharge to be accounted for by individual characteristics or other unknown factors (Table 4).

## Discussion

We evaluated the trend in the prevalence of STI related symptoms using the 2004, 2010 and 2022 TDHS data among people aged 15–49 years. Overall, there was an increase in the prevalence of genital ulcers and/or discharge with 4.3%, 5.0% and 9.3% being the prevalence in the 2004, 2010 and 2022 surveys, respectively. Older age, female, rural residents, married/living together, divorced/separated/widowed, having high number of lifetime sex partners, recent sexual activity and age of sexual debut were all associated with an increased odds of having of genital ulcers/discharge.

A consistent increase in the prevalence of genital ulcers/discharge in this setting is concerning given the association between STIs and HIV. It is a well-established fact that having STIs is associated with an increased risk of HIV acquisition and transmission [4]. STIs facilitate viral

**Table 2. Trends in the prevalence of self-reported genital ulcers/discharge among 15–49 years old participants by characteristics, from the 2004, 2010 and 2022 demographic health surveys in Tanzania.**

| Characteristics | Overall | 2004 | 2010 | 2022 | 2004–2010 | 2010–2022 | 2004–2022 |
|---|---|---|---|---|---|---|---|
| | (n = 2,670) | (n = 469) | (n = 508) | (n = 1,693) | Change in STI prevalence | | |
| | n (%) | n (%) | n (%) | n (%) | % | % | % |
| Age | | | | | | | |
| 15–24 | 767 (5.0) | 150 (3.0) | 153 (3.9) | 464 (6.8) | 0.9 | 2.9 | 3.8 |
| 25–49 | 1,903 (7.9) | 319 (5.1) | 355 (5.7) | 1,229 (10.8) | 0.6 | 5.1 | 5.7 |
| Sex | | | | | | | |
| Male | 551 (6.0) | 92 (4.3) | 90 (4.5) | 369 (7.5) | 0.2 | 3.0 | 3.2 |
| Female | 2,119 (6.9) | 377 (4.2) | 418 (5.1) | 1,324 (9.9) | 0.9 | 4.8 | 5.7 |
| Education | | | | | | | |
| No education | 457 (6.6) | 107 (4.2) | 92 (5.7) | 258 (9.7) | 1.5 | 4.0 | 5.5 |
| ≥ Primary education | 2213 (6.7) | 362 (4.3) | 416 (4.9) | 1,435 (9.2) | 0.6 | 4.3 | 4.9 |
| Marital status | | | | | | | |
| Never married | 401 (3.6) | 64 (2.0) | 65 (2.5) | 272 (5.1) | 0.5 | 2.6 | 3.1 |
| Married/living together | 1,853 (7.6) | 340 (4.9) | 368 (5.8) | 1,145 (10.5) | 0.9 | 4.7 | 5.6 |
| Divorced/separated/widowed | 416 (10.5) | 65 (5.9) | 75 (7.3) | 276 (14.8) | 1.4 | 7.5 | 8.9 |
| Residence | | | | | | | |
| Urban | 779 (6.8) | 112 (4.5) | 127 (5.1) | 540 (8.7) | 0.6 | 3.6 | 4.2 |
| Rural | 1891 (6.7) | 357 (4.2) | 381 (5.0) | 1,153 (9.6) | 0.8 | 4.6 | 5.4 |
| Wealth quantile | | | | | | | |
| Lowest | 470 (6.7) | 87 (3.9) | 102 (5.6) | 281 (9.4) | 1.7 | 3.8 | 5.5 |
| Second or higher | 2,200 (6.7) | 382 (4.3) | 406 (4.9) | 1,412 (9.2) | 0.6 | 4.3 | 4.9 |
| Tested for HIV and received results* | | | | | | | |
| No | 783 (4.2) | 394 (4.0) | 210 (4.8) | 179 (4.0) | 0.8 | -0.8 | 0.0 |
| Yes | 1,876 (9.1) | 71 (6.1) | 291 (5.3) | 1,514 (11.0) | -0.8 | 5.7 | 4.9 |
| Number of lifetime sexual partners* | | | | | | | |
| 1–2 | 957 (6.7) | Na | 250 (4.5) | 707 (8.4) | N/A | 3.9 | N/A |
| 3–4 | 603 (10.8) | Na | 146 (7.4) | 457 (12.7) | N/A | 5.3 | N/A |
| >4 | 640 (12.8) | Na | 111 (9.1) | 529 (14.2) | N/A | 5.1 | N/A |
| Number of partners/wives* | | | | | | | |
| 0 | 1,196 (7.7) | 216 (4.9) | 217 (5.2) | 763 (11.3) | 0.3 | 6.1 | 6.4 |
| 1 | 533 (7.0) | 91 (4.5) | 124 (6.7) | 318 (8.8) | 2.2 | 2.1 | 4.3 |
| >1 | 124 (8.5) | 33 (7.9) | 27 (6.6) | 64 (10.6) | -1.3 | 4.0 | 2.7 |
| Recent sexual activity | | | | | | | |
| Never had sex | 0 (0.0) | 0 (0.0) | 0 (0.0) | 0 (0.0) | | | |
| Active in the last 4 weeks | 1,868 (8.5) | 322 (5.4) | 372 (6.5) | 1174 (11.6) | 1.1 | 5.1 | 6.2 |
| Not active in the last 4 weeks | 802 (6.7) | 147 (4.0) | 136 (4.8) | 519 (9.7) | 0.8 | 4.9 | 5.7 |
| Ever heard of STI | | | | | | | |
| No | 253 (5.9) | 1 (1.7) | 0 (0.0) | 251 (6.1) | -1.7 | 6.1 | 4.4 |
| Yes | 2,326 (7.0) | 376 (4.3) | 508 (5.0) | 1442 (10.1) | 0.7 | 5.1 | 5.8 |
| Unknown | 92 (4.3) | 92 (4.3) | 0 (0.0) | 0 (0.0) | -4.3 | 0.0 | -4.3 |
| Age at first sex | | | | | | | |
| <15 | 398 (3.9) | 88 (2.7) | 78 (3.0) | 232 (5.2) | 0.3 | 2.2 | 2.5 |
| 15–19 | 1885 (8.2) | 325 (5.2) | 349 (6.2) | 1211 (11.2) | 1.0 | 5.0 | 6.0 |
| ≥20 | 387 (5.9) | 56 (3.2) | 81 (4.3) | 250 (8.7) | 1.1 | 4.4 | 5.5 |
| Number of alcohol drinks per day* | | | | | | | |
| 0–1 | 55 (13.6) | N/A | N/A | 55 (13.6) | N/A | N/A | N/A |

*(Continued)*

**Table 2.** (Continued)

| Characteristics | Overall | 2004 | 2010 | 2022 | 2004–2010 | 2010–2022 | 2004–2022 |
|---|---|---|---|---|---|---|---|
| | (n = 2,670) | (n = 469) | (n = 508) | (n = 1,693) | Change in STI prevalence | | |
| | n (%) | n (%) | n (%) | n (%) | % | % | % |
| 2–3 | 119 (13.8) | N/A | N/A | 119 (13.8) | N/A | N/A | N/A |
| ≥4 | 83 (15.6) | N/A | N/A | 83 (15.6) | N/A | N/A | N/A |
| Wife request husband to use condom if he had STI | | | | | | | |
| No | 439 (5.3) | 60 (4.2) | 58 (4.6) | 321 (5.8) | 0.4 | 1.2 | 1.6 |
| Yes | 2,061 (7.6) | 305 (4.5) | 439 (5.3) | 1,317 (11.2) | 0.8 | 5.9 | 6.7 |
| Unknown | 77 (2.8) | 12 (1.5) | 10 (1.7) | 55 (4.4) | 0.2 | 2.7 | 2.9 |
| Total | 2,670 (6.7) | 469 (4.3) | 508 (5.0) | 1,693 (9.3) | 0.7 | 4.3 | 5.0 |

Abbreviations:

*indicates number of people assessed were less than 46,481; n indicates number of people with discriminatory attitudes; HIV indicates Human immune deficiency virus; N/A indicates not applicable.

entry through genital sores [26,27]. Having STIs also increases the concentration of HIV in the genital tract, thereby increases infectiousness of people living with HIV [28]. Furthermore, blood concentration of HIV may also be increased among people living with HIV coinfected with STIs, such as gonorrhea and chlamydia [29]. The increase in the prevalence of genital ulcers/discharge in this setting needs immediate attention. Besides increased risk of HIV acquisition and transmission, STIs have a negative impact in sexual and reproductive health such as infertility[1–3]. There is a need to increase people's awareness of symptomatic STIs and their associated negative consequences. As most STIs are manageable, regular testing and treatment, partner notification, condom use and avoiding engaging in high-risk behaviors, such as having multiple sex partners, can reduce the prevalence of genital ulcers/discharge and STIs overall. Although these simple interventions have proven benefits in reducing STI acquisitions and transmissions, uptake of condom use and adherence to behavior change is low [7,30,31]. Consistent health education to increase public awareness around STIs would be beneficial. Furthermore, as with HIV, addressing stigma around other STIs may increase health seeking behavior and prevent spread of STIs. Moreover, investing in accurate diagnostic tools, such as molecular technologies and point-of-care tests, is critical. Since most patients with STIs are managed symptomatically [32], asymptomatic cases may be missed leading to persistent STIs. Improving diagnosis would not only appropriately treat those diagnosed but also improve antibiotic stewardship by not treating those who are misdiagnosed symptomatically. A review of DHS reports from other countries in Africa also indicated an increase in the symptoms of STIs. For example, the 2004, 2010 and 2015 DHS from Malawi showed that 3%, 5% and 10% of women, respectively, reported abnormal discharge and the corresponding prevalence of discharge in men was 3%, 3% and 5.2%, respectively [33–35]. In the same survey years, 6%, 8.3% and 6.4% of women and 3%, 4% and 6.5% of men, respectively, reported genital ulcers. In Kenya, the 2003, 2014 and 2022 DHS showed that 3%, 4.3% and 9.6% of women and 2%, 1.5% and 2.5% of men, respectively, reported abnormal discharge [36–38]. In those survey years, 2%, 2.4% and 5.2% of women and 2%, 1.1% and 2.5% of men, respectively, reported genital ulcers. Although there was an increased trend in the prevalence of genital ulcers/discharge in Tanzania, DHS data from Malawi, Mali and Uganda reported higher prevalence of these STI related symptoms. While overall prevalence of genital ulcers/discharge in Tanzania was nearly 7%, reported prevalence in Malawi, Mali and Uganda were 11%, 14% and 26%, respectively [18,39,40]. Addressing high prevalence of STI symptoms in the African region is crucial given the high prevalence of HIV in the region [41].

**Table 3. Bivariate analysis for factors associated with self-reported genital ulcers/discharge among 15–49 years old participants in Tanzania.**

| Characteristics | n (%) | OR (95% CI) | p-value |
|---|---|---|---|
| Age | | | |
| 15–24 | 767 (5.0) | 1.0 | |
| 25–49 | 1,903 (7.9) | 1.65 (1.48–1.83) | <.001 |
| Sex | | | |
| Male | 551 (6.0) | 1.0 | |
| Female | 2,119 (6.9) | 1.16 (1.02–1.31) | 0.021 |
| Education | | | |
| No education | 457 (6.6) | 1.0 | |
| ≥ Primary education | 2213 (6.7) | 1.01 (0.89–1.16) | 0.841 |
| Marital status | | | |
| Never married | 401 (3.6) | 1.0 | |
| Married/living together | 1,853 (7.6) | 2.16 (1.87–2.50) | <.001 |
| Divorced/separated/widowed | 416 (10.5) | 3.12 (2.62–3.71) | <.001 |
| Residence | | | |
| Urban | 779 (6.8) | 1.0 | |
| Rural | 1891 (6.7) | 0.99 (0.87–1.13) | 0.855 |
| Wealth quantile | | | |
| Lowest | 470 (6.7) | 1.0 | |
| Second or higher | 2,200 (6.7) | 0.99 (0.88–1.13) | 0.954 |
| Tested HIV & received results | | | |
| No | 783 (4.2) | 1.0 | |
| Yes | 1,876 (9.1) | 2.27 (2.02–2.56) | <.001 |
| Number of lifetime sex partners* | | | |
| 1–2 | 957 (6.7) | 1.0 | |
| 3–4 | 603 (10.8) | 1.67 (1.46–1.91) | <.001 |
| >4 | 640 (12.8) | 2.04 (1.77–2.35) | <.001 |
| Number of partners/wives* | | | |
| 0 | 1,196 (7.7) | 1.0 | |
| 1 | 533 (7.0) | 0.89 (0.78–1.03) | 0.116 |
| >1 | 124 (8.5) | 1.10 (0.87–1.40) | 0.419 |
| Recent sexual activity | | | |
| Never had sex | 0 (0.0) | N/A | |
| Active in the last 4 weeks | 1,868 (8.5) | 1.28 (1.16–1.42) | <.001 |
| Not active in the last 4weeks | 802 (6.7) | 1.0 | |
| Survey year | | | |
| 2004 | 469 (4.3) | 1.0 | |
| 2010 | 508 (5.0) | 1.19 (1.01–1.39) | 0.033 |
| 2022 | 1,693 (9.3) | 2.30 (2.00–2.64) | <.001 |
| Ever heard of STI | | | |
| No | 253 (5.9) | 1.0 | |
| Yes | 2,326 (7.0) | 1.19 (0.99–1.43) | 0.062 |
| Age at first sex | | | |
| <15 | 398 (3.9) | 1.0 | |
| 15–19 | 1885 (8.2) | 2.18 (1.90–2.50) | <.001 |
| ≥20 | 387 (5.9) | 1.53 (1.27–1.84) | <.001 |
| Number of alcohol drinks per day* | | | |

*(Continued)*

**Table 3.** (Continued)

| Characteristics | n (%) | OR (95% CI) | p-value |
|---|---|---|---|
| 0–1 | 55 (13.6) | 1.0 | |
| 2–3 | 119 (13.8) | 1.02 (0.69–1.50) | 0.936 |
| ≥4 | 83 (15.6) | 1.18 (0.76–1.82) | 0.470 |
| Wife request husband to use condom if he had STI | | | |
| No | 439 (5.3) | 1.0 | |
| Yes | 2,061 (7.6) | 1.50 (1.28–1.73) | <.001 |

Abbreviations: n indicates number of people; HIV indicates human immune deficiency virus; OR indicates odds ratio; CI indicates confidence interval; N/A indicates not applicable; STI indicates sexually transmitted infection

We noted higher odds of genital ulcers/discharge among divorced/separated or widowed individuals compared to singles. When a subgroup analysis of this association was conducted, the association between divorced/separated or widowed individuals and genital ulcers/discharge was statistically significant for women but not for men. As we have also seen higher odds of genital ulcers/discharge among women compared to men, this association is not surprising due the matters related to biologic nature of women [42] and gender inequality characterized by inferiority with condom use negotiations, which can cause women to have a higher risk of acquiring STIs compared to men [43,44]. It is worth to note that women are more likely than men to have non STI genital ulcers/discharge such as vaginal thrush, bacterial vaginosis, allergies.

We found that respondents who reported having had first sex at the age younger than 15 years had higher risk of reporting genital ulcers/discharge compared to those who had first sex when they were 20 years or older. These findings are similar to other studies that showed higher risk of STI for individuals who had first sex at younger age [45,46]. Interventions aimed at delaying initiation of sexual intercourse and provision of health education to young individuals prior to the engagement of sexual activities are critical to prevent future development of STI and associated symptoms.

The main strength of this analysis is the use of nationally representative data from three consecutive surveys that enabled us to explore the trend of the prevalence of genital ulcers/discharge in Tanzania. We recognize a few limitations of this analysis. Although our findings showed a consistent upward trend in the prevalence of genital ulcers/discharges, we were unable to determine the specific underlying causes with certainty. It is possible that factors such as decreased stigma surrounding sexually transmitted infections (STIs) may have encouraged more individuals to report symptoms and seek care. Additionally, changes in STI screening policies could have led to the identification of more cases. However, due to the lack of direct data linking these potential causes with our observations, this remains speculative. This limitation restricts our ability to provide a more definitive explanation for the observed trends. Due to the inherent recall bias associated with the cross-sectional nature of the surveys, it is likely that the prevalence of symptomatic STIs may be underreported, and underreporting could also be due to the use of symptoms to quantify STIs, excluding asymptomatic STIs. Additionally, as this analysis focuses on self-reported symptoms, asymptomatic STIs, which contribute significantly to transmission, may not be captured, underscoring the need for broader screening efforts in future studies. Despite these limitations, self-reported data have been widely used in different studies, often with high sensitivity, specificity and agreement when compared with gold standard measures [47,48] and associations reported in this study were plausible.

**Table 4. Multivariable analysis for factors associated with self-reported genital ulcers/discharge among 15–49 years old participants in Tanzania.**

| | Model 1 | | Model 2 | | Model 3 | |
|---|---|---|---|---|---|---|
| | aOR (95% CI) | p-value | aOR (95% CI) | p-value | aOR (95% CI) | p-value |
| **Fixed effects** | | | | | | |
| Intercept, β(SE) | -2.96 (0.22) | 0.005 | -2.98 (0.36) | 0.077 | -2.94 (0.21) | <.001 |
| Age | | | | | | |
| 15–24 | | | 1.0 | | 1.0 | |
| 25–49 | | | 1.19 (1.06–1.33) | 0.002 | 1.19 (1.05–1.33) | 0.002 |
| Sex | | | | | | |
| Male | | | 1.0 | | 1.0 | |
| Female | | | 1.68 (1.48–1.90) | <.001 | 1.68 (1.48–1.90) | <.001 |
| Education | | | | | | |
| No education | | | 1.0 | | 1.0 | |
| ≥ Primary education | | | 1.08 (0.94–1.23) | 0.277 | 1.08 (0.94–1.23) | 0.279 |
| Marital status | | | | | | |
| Never married | | | 1.0 | | 1.0 | |
| Married/living together | | | 0.96 (0.84–1.12) | 0.637 | 0.96 (0.83–1.12) | 0.644 |
| Divorced/separated/widowed | | | 1.17 (0.98–1.40) | 0.089 | 1.17 (1.01–1.40) | 0.048 |
| Residence | | | | | | |
| Urban | | | 1.0 | | 1.0 | |
| Rural | | | 1.12 (0.96–1.29) | 0.145 | 1.05 (0.92–1.19) | 0.485 |
| Wealth quantile | | | | | | |
| Lowest | | | 1.0 | | 1.0 | |
| Second or higher | | | 0.89 (0.78–1.02) | 0.083 | 0.89 (0.79–1.02) | 0.083 |
| Age at first sex | | | | | | |
| <15 | | | 1.0 | | 1.0 | |
| 15–19 | | | 1.02 (0.90–1.17) | 0.747 | 1.02 (0.89–1.17) | 0.750 |
| ≥20 | | | 0.89 (0.74–1.09) | 0.178 | 0.85 (0.74–0.99) | 0.046 |
| Number of lifetime sex partners* | | | | | | |
| 1–2 | | | 1.0 | | 1.0 | |
| 3–4 | | | 1.61 (1.44–1.80) | <.001 | 1.61 (1.44–1.80) | <.001 |
| >4 | | | 2.20 (1.94–2.50) | <.001 | 2.20 (1.94–2.50) | <.001 |
| Recent sexual activity in the last 4 weeks | | | | | | |
| Never had sex | | | N/A | | N/A | |
| Active | | | 1.28 (1.14–1.43) | <.001 | 1.28 (1.14–1.43) | <.001 |
| Not active | | | 1.0 | | 1.0 | |
| Wife may request husband to use condom if he had STI | | | | | | |
| No | | | 1.0 | | 1.0 | |
| Yes | | | 1.77 (1.57–2.00) | <.001 | 1.77 (1.57–2.00) | <.001 |
| Ever heard of STI | | | | | | |
| No | | | 1.0 | | 1.0 | |
| Yes | | | 1.17 (1.00–1.36) | 0.046 | 1.17 (1.00–1.36) | 0.043 |
| Tested HIV & received results | | | | | | |
| No | | | 1.0 | | 1.0 | |
| Yes | | | 0.98 (87–1.18) | 0.794 | 0.98 (87–1.18) | 0.794 |
| Survey year | | | | | | |
| 2004 | | | | | 1.0 | |
| 2010 | | | | | 1.11 (0.98–1.26) | 0.110 |
| 2022 | | | | | 2.27 (2.00–2.52) | <.001 |

*(Continued)*

**Table 4.** (Continued)

| | Model 1 | | Model 2 | | Model 3 | |
|---|---|---|---|---|---|---|
| **Random effects** | | | | | | |
| Error variance – Level 2 intercept, β(SE) | 0.14 (0.12) | 0.112 | 0.16 (0.16) | 0.161 | 0.17 (0.15) | |
| ICC | 0.04 | | 0.05 | | 0.05 | |
| **Model fit** | | | | | | |
| AIC | 20070.28 | | 14226.92 | | 14215.13 | |
| BIC | 20068.48 | | 14204.70 | | 14192.91 | |
| -2LL | 20066.28 | | 14192.92 | | 14181.13 | |
| Assessments | N/A | | Model 2 vs Model 1 | | Model 3 vs Model 2 | |
| P-value | N/A | | <.001 | | <.001 | |

Abbreviations: STI indicates sexually transmitted infections; HIV indicates Human immune deficiency virus; aOR indicates adjusted odds ratio; CI indicates confidence intervals; N/A indicates not applicable, ICC indicates intra class correlation coefficient; β indicates regression coefficient; SE indicates standard error; AIC indicates Akaike information criterion; BIC indicates Bayesian information criterion.

## Conclusions

There was an increase in the trend of prevalence of genital ulcers/discharge in Tanzania with women and divorced/separated and widowed individuals being disproportionally affected. Identifying reasons for the increase in these STI related symptoms is paramount to strategize and address challenges. Empowering women through condom negotiation skills, partner discussions about safe sex practices and regular STI testing are critical to minimize the prevalence of genital ulcers/discharge. Emphasizing behavior change interventions along with the incorporation of assisted partner notification services at STI clinics would also be beneficial. Strengthening sex education programs, particularly in rural areas, and expanding the use of mobile health platforms for STI testing, will be essential in curbing the rising trend of symptomatic STIs.

## Author contributions

**Conceptualization:** Zuhura Mbwana Ally, Lynn Moshi, Hafidha Mhando Bakari, Habib Omari Ramadhani.

**Data curation:** Rahma Musoke, Mariam Salim Mbwana, Leticia Francis Karia.

**Formal analysis:** Haji Mbwana Ally, Hassan Fredrick Fussi, Habib Omari Ramadhani.

**Investigation:** Lynn Moshi, Hafidha Mhando Bakari, Glenda Marie Manayon.

**Methodology:** Hassan Fredrick Fussi, Habib Omari Ramadhani.

**Software:** Rahma Musoke.

**Supervision:** Habib Omari Ramadhani.

**Validation:** Mariam Salim Mbwana, Hafidha Mhando Bakari, Swalehe Mustafa Salim.

**Visualization:** Rahma Musoke, Maximillian Francis Karia, Alpha Johnson Kapola.

**Writing – original draft:** Zuhura Mbwana Ally.

**Writing – review & editing:** Leticia Francis Karia, Alpha Johnson Kapola, Glenda Marie Manayon, Hassan Fredrick Fussi, Habib Omari Ramadhani.

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
