## [Decision Letter · Decision Letter 0]

27 Aug 2024

PONE-D-24-28121Trend of sexually transmitted infections among survey respondents in Tanzania: Analysis of demographic health surveys.PLOS ONE

Dear Dr. Ally,

Thank you for submitting your manuscript to PLOS ONE. After careful consideration, we feel that it has merit but does not fully meet PLOS ONE’s publication criteria as it currently stands. Therefore, we invite you to submit a revised version of the manuscript that addresses the points raised during the review process.

We look forward to receiving your revised manuscript.

Kind regards,

Yimam Getaneh Misganie (PhD, PhD)

Academic Editor

PLOS ONE

Journal Requirements: When submitting your revision, we need you to address these additional requirements. 1. Please ensure that your manuscript meets PLOS ONE's style requirements, including those for file naming. The PLOS ONE style templates can be found at https://journals.plos.org/plosone/s/file?id=wjVg/PLOSOne_formatting_sample_main_body.pdf and https://journals.plos.org/plosone/s/file?id=ba62/PLOSOne_formatting_sample_title_authors_affiliations.pdf 2. Please include a complete copy of PLOS’ questionnaire on inclusivity in global research in your revised manuscript. Our policy for research in this area aims to improve transparency in the reporting of research performed outside of researchers’ own country or community. The policy applies to researchers who have travelled to a different country to conduct research, research with Indigenous populations or their lands, and research on cultural artefacts. The questionnaire can also be requested at the journal’s discretion for any other submissions, even if these conditions are not met.  Please find more information on the policy and a link to download a blank copy of the questionnaire here: https://journals.plos.org/plosone/s/best-practices-in-research-reporting. Please upload a completed version of your questionnaire as Supporting Information when you resubmit your manuscript. 3. We note that your Data Availability Statement is currently as follows: All relevant data are within the manuscript and its Supporting Information files. Please confirm at this time whether or not your submission contains all raw data required to replicate the results of your study. Authors must share the “minimal data set” for their submission. PLOS defines the minimal data set to consist of the data required to replicate all study findings reported in the article, as well as related metadata and methods (https://journals.plos.org/plosone/s/data-availability#loc-minimal-data-set-definition). For example, authors should submit the following data: - The values behind the means, standard deviations and other measures reported;- The values used to build graphs;- The points extracted from images for analysis. Authors do not need to submit their entire data set if only a portion of the data was used in the reported study. If your submission does not contain these data, please either upload them as Supporting Information files or deposit them to a stable, public repository and provide us with the relevant URLs, DOIs, or accession numbers. For a list of recommended repositories, please see https://journals.plos.org/plosone/s/recommended-repositories. If there are ethical or legal restrictions on sharing a de-identified data set, please explain them in detail (e.g., data contain potentially sensitive information, data are owned by a third-party organization, etc.) and who has imposed them (e.g., an ethics committee). Please also provide contact information for a data access committee, ethics committee, or other institutional body to which data requests may be sent. If data are owned by a third party, please indicate how others may request data access. 4. Please amend your manuscript to include your abstract after the title page.

Reviewers' comments:

**Comments to the Author**

1. Is the manuscript technically sound, and do the data support the conclusions?

Reviewer #1: Partly

Reviewer #2: Partly

2. Has the statistical analysis been performed appropriately and rigorously? 

Reviewer #1: Yes

Reviewer #2: No

3. Have the authors made all data underlying the findings in their manuscript fully available?

Reviewer #1: Yes

Reviewer #2: Yes

4. Is the manuscript presented in an intelligible fashion and written in standard English?

Reviewer #1: Yes

Reviewer #2: Yes

5. Review Comments to the Author

Reviewer #1: Summary

The authors evaluated trends in the prevalence of STIs using the 2004, 2010 and 2022 Tanzania using Demographic Health Survey (TDHS) data. Participants aged 15 to 49 years who reported having had genital ulcers and/or discharge within 12 months prior to the survey were included in the analysis.

General Comment

The manuscript is clearly written and addresses an important topic.

It could however benefit from some edits.

Title: This analysis only evaluated data for respondents who had genital ulcers and/or discharge, not all STIs in general. Therefore, consider editing the title to read as follows; Trend of genital ulcers and genital discharge respondents in Tanzania: Analysis of demographic health surveys.

Also consider editing the conclusion to convey that the reported increase in prevalence was for ulcers and/or discharge, not all symptomatic STIs. Symptoms of STIs go beyond these two.

Variable definition:

Line 104-105: It is stated that a person was considered to ever had STI’s if he/she reported the presence of genital ulcers and/or discharge. Time dimension is missing in this definition. Please edit to improve clarity.

Discussion:

Ensure that the range of STIs covered in the studies referenced for comparison was the same as for TDHS (i.e. genital ulcers and/or discharge).

Ensure that all recommendations are supported by the findings of the study.

Reviewer #2: Major considerations

• Why only the three TDHS reports were included as far as other data sets are available? or is there a specific reason for not including other available datasets? It would be helpful to explicitly mention the reason, otherwise all available dataset should be included to have a reliable trend analysis.

• Factors contributing for trends of STI should be highlighted on the title as well as introduction part for consistency. Identified factors associated with STIs globally and regionally from previous report should be mentioned in the introduction part, too. Better to introduce the importance of understanding not just only the prevalence but also the factors contributing to STIs to inform targeted interventions.

• DHS survey is a hierarchical or nested data (respondents are nested from clusters), suggesting the presence intra-group correlations. Using ordinary regression analysis brings a biased estimate, hence hierarchical analysis models should be utilized, as they account for clustering effect.

• According to the TDHS 2022 data (as of the 2022 Demographic and Health Survey and Malaria Indicator Survey report), “14% of women and 13% of men aged 15–49 years reported having an STI or symptoms of an STI in the 12 months preceding the survey” However, the author reported a prevalence of STIs of 7.5% in men and 9.9% in women (Table 2). How should this discrepancy be addressed, or is there any justification the authors considered? Similarly, a discrepancy for other TDHS reports.

Minor comments

- [ Title]: Given that the information in study is based on self-reported data and does not reflect confirmed diagnoses of STIs, let modify the title. Also consider including the time frame in the title. (e.g., “Trends in Self-reported Sexually Transmitted Infections among Tanzanian Survey Respondents, 2004-2022: Analysis of Demographic Health Surveys”).

- [Abstract]- Using abbreviation is not recommended

- [ Introduction]: the introduction could benefit from a more structured flow. Consider separating the global context from the regional context to improve readability.

- [ Result]: Factors reported in the TDHS such as circumcision, women’s justification for using a condom, awareness of STIs, substance use, and others that may potentially be associated with STIs, should also be considered in the regression analysis. Including these variables could provide a more comprehensive understanding of the factors influencing STI prevalence and help identify additional significant predictors.

- [ Discussion]:

o It would be beneficial to provide more context on why the increased trend might have occurred, especially between 2010 and 2022. Are there known changes in behavior, policy, or healthcare access during this period that could explain the trends?

o The discussion could also benefit from a comparison with findings from other countries or regions. How do the trends in Tanzania compare with other parts of sub-Saharan Africa or globally? This would help position the findings within a broader context

o A significant association of STIs with some predictor variables has been reported. The author would benefit from explaining the implications and justifications for these associations.

- [Conclusion]: could be more action-oriented, suggest specific interventions that could help address the rising trend of STIs.

Line [71] “Moreover, STIs increases risk of cancer (HPV and cervical/anal/head and neck cancer) (5,6)”. How STIs increase the risk of head and neck cancers? evidence? even the provided references not support the statement.

Line 77-78: “Prospectively collected data from eight health care facilities in Tanzania”……What about the consecutive TDHZ reports?

Line 101-102: How participants who were unsure about their STI status were handled beyond just exclusion?

Line 105-106: The rational for categorization of some variables (e.g. age) could be better justified and cited.

Line 120-122: Criteria to included variables for the final multivariable model is not clear. If the number of variables is large enough, better to use a statistical criterion (often using a p-value threshold, p < 0.25 or < 0.20) to reduce the number of variables considered, otherwise all variables should be included in the final multivariable regression analysis.

- Including model fitness tests and model assumption tests in the statistical analysis section is crucial for ensuring the validity of the result.

6. PLOS authors have the option to publish the peer review history of their article (what does this mean? ). If published, this will include your full peer review and any attached files.

**Do you want your identity to be public for this peer review?** For information about this choice, including consent withdrawal, please see our Privacy Policy .

Reviewer #1: No

Reviewer #2: No

---

## [Author Response · Author response to Decision Letter 1]

2 Oct 2024

Dear editors PLOS ONE journal,

This letter serves as response to reviewer’s comments for the manuscript “Trend of genital ulcers and discharge and associated factors sexually transmitted infections among survey respondents in Tanzania, 2004-2022: Analysis of demographic health surveys.” We greatly appreciate editors and reviewers and believe their comments have significantly improved our manuscript.

Below are point by point responses

1. Please ensure that your manuscript meets PLOS ONE's style requirements

The manuscript meets PLOS ONE’s style requirements

2. Please include a complete copy of PLOS’ questionnaire

a. A copy of PLOS questionnaire has been provided

3. Data Availability Statement

a. DHS data is publicly available at DHS website upon request. We have provided a link to where raw dataset can be requested and downloaded.

4. Please amend your manuscript to include your abstract after the title page

a. We have included abstract after tittle page.

Reviewer #1: Summary

The authors evaluated trends in the prevalence of STIs using the 2004, 2010 and 2022 Tanzania using Demographic Health Survey (TDHS) data. Participants aged 15 to 49 years who reported having had genital ulcers and/or discharge within 12 months prior to the survey were included in the analysis.

General Comment

The manuscript is clearly written and addresses an important topic.

It could however benefit from some edits.

Title: This analysis only evaluated data for respondents who had genital ulcers and/or discharge, not all STIs in general. Therefore, consider editing the title to read as follows; Trend of genital ulcers and genital discharge respondents in Tanzania: Analysis of demographic health surveys.

We thank you for your comments. We have made the changes as advised.

Also consider editing the conclusion to convey that the reported increase in prevalence was for ulcers and/or discharge, not all symptomatic STIs. Symptoms of STIs go beyond these two.

We thank you for your comments. We have made the changes as advised.

Variable definition:

Line 104-105: It is stated that a person was considered to ever had STI’s if he/she reported the presence of genital ulcers and/or discharge. Time dimension is missing in this definition. Please edit to improve clarity.

We thank you for your comments. We have included time frame to improve clarity.

Discussion:

Ensure that the range of STIs covered in the studies referenced for comparison was the same as for TDHS (i.e. genital ulcers and/or discharge).

We thank you for your comments. Where we made direct comparisons, all references included were related to the same symptoms used for our study. Moreover, comparison were made using demographic health surveys data from different countries such as Uganda, Kenya, Mali and Malawi.

Ensure that all recommendations are supported by the findings of the study.

We thank you for your comments. We have taken note and edit accordingly.

Reviewer #2: Major considerations

• Why only the three TDHS reports were included as far as other data sets are available? or is there a specific reason for not including other available datasets? It would be helpful to explicitly mention the reason, otherwise all available dataset should be included to have a reliable trend analysis.

We thank you for your comments. DHS are usually conducted every 5 years. However, the 2015/2016 TDHS publicly available dataset does not contain variables related to abnormal discharge and genital ulcers. We agree having more data would improve trend analysis, however, we are limited with the availability of the variables in the shared datasets. We have added this information in the methods.

• Factors contributing for trends of STI should be highlighted on the title as well as introduction part for consistency. Identified factors associated with STIs globally and regionally from previous report should be mentioned in the introduction part, too. Better to introduce the importance of understanding not just only the prevalence but also the factors contributing to STIs to inform targeted interventions.

We thank you for your comments. We have provided details for factors associated with increased STI risk and narrated the importance of further exploration.

• DHS survey is a hierarchical or nested data (respondents are nested from clusters), suggesting the presence intra-group correlations. Using ordinary regression analysis brings a biased estimate, hence hierarchical analysis models should be utilized, as they account for clustering effect.

We thank you for your comments. Initial analysis did not use ordinary logistic regression, but regression models used to analyze survey datasets that accounted for survey weights and clustering. We have added hierarchical multivariable models and introduced a new table 4 in the revised manuscript.

• According to the TDHS 2022 data (as of the 2022 Demographic and Health Survey and Malaria Indicator Survey report), “14% of women and 13% of men aged 15–49 years reported having an STI or symptoms of an STI in the 12 months preceding the survey” However, the author reported a prevalence of STIs of 7.5% in men and 9.9% in women (Table 2). How should this discrepancy be addressed, or is there any justification the authors considered? Similarly, a discrepancy for other TDHS reports.

We thank you for your comments. TDHS had three questions related to STI. (Had any STI in last 12 months; Had genital sore/ulcer in last 12 months and Had genital discharge in last 12 months). The interest for this analysis was to look for symptomatology and focused on the last two questions. The discrepancy in the reporting is likely due to the exclusion of the first question that generally asked presence of STI without specific symptoms. We made changes throughout the manuscript emphasizing genital ulcers/discharge.

Minor comments

- [ Title]: Given that the information in study is based on self-reported data and does not reflect confirmed diagnoses of STIs, let modify the title. Also consider including the time frame in the title. (e.g., “Trends in Self-reported Sexually Transmitted Infections among Tanzanian Survey Respondents, 2004-2022: Analysis of Demographic Health Surveys”).

We thank you for your comments: We made edits to the title considering comments from both reviewers.

- [Abstract]- Using abbreviation is not recommended.

We thank you for your comments. Due to the word limit, we are inclined to include a few common abbreviations. We have deleted uncommon abbreviations from the abstract.

- [ Introduction]: the introduction could benefit from a more structured flow. Consider separating the global context from the regional context to improve readability.

We thank you for your comments. We have made two separate paragraphs. First providing global context, then regional context as advised.

- [ Result]: Factors reported in the TDHS such as circumcision, women’s justification for using a condom, awareness of STIs, substance use, and others that may potentially be associated with STIs, should also be considered in the regression analysis. Including these variables could provide a more comprehensive understanding of the factors influencing STI prevalence and help identify additional significant predictors.

We thank you for your comments, we have added a few more variables as advised.

- [ Discussion]:

o It would be beneficial to provide more context on why the increased trend might have occurred, especially between 2010 and 2022. Are there known changes in behavior, policy, or healthcare access during this period that could explain the trends?

We thank you for your comments. These are critical questions; however, we may not be able to explain with certainty the cause of increments of these symptomatic STI. We have added a description of these findings.

o The discussion could also benefit from a comparison with findings from other countries or regions. How do the trends in Tanzania compare with other parts of sub-Saharan Africa or globally? This would help position the findings within a broader context.

We thank you for your comments. We have reviewed DHS reports of other countries and provided prevalences of symptomatic STIs.

o A significant association of STIs with some predictor variables has been reported. The author would benefit from explaining the implications and justifications for these associations.

We thank you for your comments. We have provided implications for some variables. For example, integration of HIV and STI services to identify new infections.

- [Conclusion]: could be more action-oriented, suggest specific interventions that could help address the rising trend of STIs.

We thank you for your comments. We have made edits at this section.

Line [71] “Moreover, STIs increases risk of cancer (HPV and cervical/anal/head and neck cancer) (5,6)”. How STIs increase the risk of head and neck cancers? evidence? even the provided references not support the statement.

We thank you for your comments. This was in reference to Human papilloma viruses which are known to cause several cancers including head and neck. We have deleted this information to minimize confusion.

Line 77-78: “Prospectively collected data from eight health care facilities in Tanzania”……What about the consecutive TDHZ reports?

We thank you for your comments. We have added data for two consecutive surveys at this section.

Line 101-102: How participants who were unsure about their STI status were handled beyond just exclusion?

We thank you for your comments. 166 participants who were unsure of having symptoms of STI accounted for 0.4% of the total sample size. We do not anticipate their inclusion would have significant impact. Furthermore, exploration of their demographic characteristics showed similar distributions to those included. For example, median age 26 vs 27 years, rural residence 67% vs 68% among those excluded vs those included in the analysis.

Line 105-106: The rational for categorization of some variables (e.g. age) could be better justified and cited.

We thank you for your comments. We justified categorization of some variables and made citations where appropriate

Line 120-122: Criteria to included variables for the final multivariable model is not clear. If the number of variables is large enough, better to use a statistical criterion (often using a p-value threshold, p < 0.25 or < 0.20) to reduce the number of variables considered, otherwise all variables should be included in the final multivariable regression analysis.

We thank you for your comments. We have made changes in this section.

- Including model fitness tests and model assumption tests in the statistical analysis section is crucial for ensuring the validity of the result.

We thank you for your comments, we have included how we assessed fitness of the models in the analysis section.

---

## [Decision Letter · Decision Letter 1]

10 Oct 2024

PONE-D-24-28121R1Trend of genital ulcers and discharge and associated factors among survey respondents in Tanzania, 2004-2022: Analysis of demographic health surveys.PLOS ONE

Dear Dr. Ally,

Thank you for submitting your manuscript to PLOS ONE. After careful consideration, we feel that it has merit but does not fully meet PLOS ONE’s publication criteria as it currently stands. Therefore, we invite you to submit a revised version of the manuscript that addresses the points raised during the review process.

We look forward to receiving your revised manuscript.

Kind regards,

Yimam Getaneh (MSc, PhD, PhD)

Academic Editor

PLOS ONE

Reviewer's Responses to Questions

**Comments to the Author**

1. If the authors have adequately addressed your comments raised in a previous round of review and you feel that this manuscript is now acceptable for publication, you may indicate that here to bypass the “Comments to the Author” section, enter your conflict of interest statement in the “Confidential to Editor” section, and submit your "Accept" recommendation.

Reviewer #1: All comments have been addressed

Reviewer #2: (No Response)

2. Is the manuscript technically sound, and do the data support the conclusions?

Reviewer #1: Yes

Reviewer #2: Yes

3. Has the statistical analysis been performed appropriately and rigorously? 

Reviewer #1: Yes

Reviewer #2: No

4. Have the authors made all data underlying the findings in their manuscript fully available?

Reviewer #1: Yes

Reviewer #2: Yes

5. Is the manuscript presented in an intelligible fashion and written in standard English?

Reviewer #1: Yes

Reviewer #2: No

6. Review Comments to the Author

Reviewer #1: The authors have addressed my comments satisfactorily in their response. I also noted satisfactory response to crosscutting comments raised by the second reviewer and me.

Reviewer #2: Major concern,

My major concern regarding the statistical analysis remains unresolved. The hierarchical models as the authors present in the revised manuscript (Model 1: Survey year, Model 2: Demographic characteristics + Survey year, Model 3: Risk behavior + other characteristics) lack clarity and do not align with the appropriate methodology for context of the study. For a better analysis in this type of setting, they should consider appropriate hierarchical models like multilevel regression analysis to account for the hierarchical nature of the data(i.e it accounts for both individual and community-level variations). Its assumption should be checked and if model assumption is satisfied, the following steps are recommended

Model 1 (Intercept-only model)- This should be the baseline model to identify whether there is significant variability between communities and individuals in the outcomes of interest (genital ulcers and discharge).

Model 2: Model that includes only individual-level characteristics (e g. age, sex…

Model 3: Model that introduces community-level variables (eg. Like Residence, community education etc. …..).

Model 4: A full model that includes both individual- and community-level characteristics.

The authors should rigorously check the assumptions of multilevel regression and use appropriate model selection criteria like AIC/BIC/PCV to determine the best-fitting model. In doing so, it could assess how much variance is explained at different levels and select the best-fitting model for final interpretation. Without proper justification and methodological rigor, the current analysis is insufficient to have accurate and interpretable results regarding the associated factors. Hence, a thorough revision of the statistical approach is essential for addressing this major concern.

7. PLOS authors have the option to publish the peer review history of their article (what does this mean? ). If published, this will include your full peer review and any attached files.

**Do you want your identity to be public for this peer review?** For information about this choice, including consent withdrawal, please see our Privacy Policy .

Reviewer #1: No

Reviewer #2: No

---

## [Author Response · Author response to Decision Letter 2]

27 Oct 2024

Dear editors PLOS ONE journal,

This letter serves as response to reviewer’s comments for the manuscript “Trend of genital ulcers and discharge and associated factors among survey respondents in Tanzania, 2004-2022: Analysis of demographic health surveys.” We greatly appreciate editors and reviewers and believe their comments have significantly improved our manuscript.

Below are point by point responses

Reviewer #1: The authors have addressed my comments satisfactorily in their response. I also noted satisfactory response to crosscutting comments raised by the second reviewer and me.

We are grateful that we responded your earlier comments.

Reviewer #2: Major concern,

My major concern regarding the statistical analysis remains unresolved. The hierarchical models as the authors present in the revised manuscript (Model 1: Survey year, Model 2: Demographic characteristics + Survey year, Model 3: Risk behavior + other characteristics) lack clarity and do not align with the appropriate methodology for context of the study. For a better analysis in this type of setting, they should consider appropriate hierarchical models like multilevel regression analysis to account for the hierarchical nature of the data(i.e it accounts for both individual and community-level variations). Its assumption should be checked and if model assumption is satisfied, the following steps are recommended

Model 1 (Intercept-only model)- This should be the baseline model to identify whether there is significant variability between communities and individuals in the outcomes of interest (genital ulcers and discharge).

Model 2: Model that includes only individual-level characteristics (e g. age, sex…

Model 3: Model that introduces community-level variables (eg. Like Residence, community education etc. …..).

Model 4: A full model that includes both individual- and community-level characteristics.

The authors should rigorously check the assumptions of multilevel regression and use appropriate model selection criteria like AIC/BIC/PCV to determine the best-fitting model. In doing so, it could assess how much variance is explained at different levels and select the best-fitting model for final interpretation. Without proper justification and methodological rigor, the current analysis is insufficient to have accurate and interpretable results regarding the associated factors. Hence, a thorough revision of the statistical approach is essential for addressing this major concern.

We thank you for your comments. We needed clarification on which variables were considered community vs individual characteristics. Based on the information provided. It is only survey year that is categorized as community level variable. For this analysis we used the terms level-1 and level-2 variables for individual characteristics and survey year respectively. The dataset was analyzed as suggested and conducted three hierarchical models as opposed to four since having a model with survey year alone would not make these models hierarchical.

---

## [Decision Letter · Decision Letter 2]

31 Jan 2025

PONE-D-24-28121R2Trend of genital ulcers and discharge and associated factors among survey respondents in Tanzania, 2004-2022: Analysis of demographic health surveys.PLOS ONE

Dear Dr. Ally,

Thank you for submitting your manuscript to PLOS ONE. After careful consideration, we feel that it has merit but does not fully meet PLOS ONE’s publication criteria as it currently stands. Therefore, we invite you to submit a revised version of the manuscript that addresses the points raised during the review process.

We look forward to receiving your revised manuscript.

Kind regards,

Joel Msafiri Francis, MD, MS, PhD

Academic Editor

PLOS ONE

Reviewers' comments:

Reviewer's Responses to Questions

**Comments to the Author**

1. If the authors have adequately addressed your comments raised in a previous round of review and you feel that this manuscript is now acceptable for publication, you may indicate that here to bypass the “Comments to the Author” section, enter your conflict of interest statement in the “Confidential to Editor” section, and submit your "Accept" recommendation.

Reviewer #1: All comments have been addressed

Reviewer #2: All comments have been addressed

Reviewer #3: (No Response)

2. Is the manuscript technically sound, and do the data support the conclusions?

Reviewer #1: Yes

Reviewer #2: Yes

Reviewer #3: Yes

3. Has the statistical analysis been performed appropriately and rigorously? 

Reviewer #1: I Don't Know

Reviewer #2: No

Reviewer #3: Yes

4. Have the authors made all data underlying the findings in their manuscript fully available?

Reviewer #1: Yes

Reviewer #2: Yes

Reviewer #3: No

5. Is the manuscript presented in an intelligible fashion and written in standard English?

Reviewer #1: Yes

Reviewer #2: Yes

Reviewer #3: Yes

6. Review Comments to the Author

Reviewer #1: All my review comments have been addressed. But the second reviewer raised substantial statistical issues that the authors need to address fully before the manuscript is published.

Reviewer #2: Upon evaluating the revised manuscript, I find that the authors have tried to addressed my concerns and and I have no further major issues to raise at this stag.

I support the acceptance of this manuscript for further processing.

Reviewer #3: Thank you for the opportunity to review this manuscript. It was revision 2 of the original manuscript but I was reviewing it for the first time. The manuscript addresses an important area in public health and was generally well written and reads easily. The authors have addressed most comments raised in previous reviews. However there a few issues that the authors can address to strengthen the manuscript and the take away messages. The main one being the issue of STI prevalence vs prevalence of genital ulcer/ discharge. Most STIs are asymptomatic and what is symptomatic is 30- 50% of the STIs. Also not everyone who has STI-associated symptoms has an STI. This is particularly the case for women and individuals with genital ulcers. A recent systematic review of causes of STI syndromes reported that (Michalow J, Walters MK, Edun O, Wybrant M, Davies B, Kufa T, Mathega T, Chabata ST, Cowan FM, Cori A, Boily MC. Aetiology of vaginal discharge, urethral discharge, and genital ulcer in sub-Saharan Africa: A systematic review and meta-regression. PLoS Medicine. 2024 May 20;21(5): e1004385) reported that 10% of men with discharge, 25% of women with a discharge and 34% of individuals with ulcers did not have an associated organism detected. The point I am making is there is need to distinguish between STI symptoms and infection as they are not always the same and the introduction and discussion needs to better reflect this and be more nuanced.

I have made specific comments

Abstract

• Line 42- its better to refer to factors associated with genital ulcers/discharge in the past 12m as opposed to STIs. I think one of the reviewers recommended a change in title but I think it should go beyond the title

• Line 45- refer to prevalence of ulcers/discharge

Introduction

• Line 78- state whether the study of STI in the 8 health facilities included people who were symptomatic or asymptomatic

• Line 79- did the 2004 Tanzanian DHS test for STIs?

• Line 82- add something about how STIs are managed in Tanzania. If its syndromic management, add information about its accuracy in predicting the presence of an STI among people with genital discharge or ulcers.

Methods

• Line 114- might be more accurate to refer to the outcome as prevalence of ulcers/ discharge OR prevalence of STI-related symptoms. This is because the presence of an ulcer or discharge does not equate to an STI particularly for women or those with ulcers.

• Line 132 - the authors dont say what variables were used for weighting

Results

• Line 175- prevalence of genital ulcer/discharge should be outcome and not STI prevalence. Please correct the results section

• Line 199- 200- Add to the discussion why you found this? Generally STIs are more prevalenct in younger people but are more likely to be asymptomatic in that group. Prevalence of symptoms and not infection increases with age as people acquire immunity

• Line 216 - women are more likely to have non-STI causes of ulcers/ discharge compared to males males eg vaginal thrush, bacterial vaginosis, allergies etc. Although women are also more likely to have prevalent STI infections as well, it will not to to the same extent as the likelihood of having ulcers/ discharge- Another systematic review showed steady burden of infections in SSA with the exception of chlamydia. (see Michalow, Julia, Hall, Lauren, Rowley, Jane, Anderson, Rebecca, Hayre, Quinton, Chico, R. Matthew, Edun, Olanrewaju, Knight, Jesse, Kuchukhidze, Salome, Majaya, Evidence, Reed, Domonique, Stevens, Oliver, Walters, Magdalene, Peters, Remco, Cori, Anne, Boily, Marie-Claude and Imai-Eaton, Jeffrey DOI: 10.1101/2024.12.16.24319070).

Discussion

• Lines 270- 277- increasing prevalence of symptoms does not imply increase in STI infection. This part of the discussion needs to be more nuanced. The two systematic reviews quoted show that the prevalence of specific STIs hasnt changed over time (except for chlamydia) among both symptomatics and aymptomatics. So unless there was testing in the surveys, there is no sufficient evidence of increase in infection but of increasing sysmptoms

• Line 278- but receiving an HIV result was not statistically significant in your adjusted model? Is there testing for STIs in Tanzania? A possible explanatiom for this may be that an HIV negative result is associated with continued high risk behaviour or doesnt change existing high risk behaviours hence the prevalence of STI-associated symptoms.

7. PLOS authors have the option to publish the peer review history of their article (what does this mean? ). If published, this will include your full peer review and any attached files.

**Do you want your identity to be public for this peer review?** For information about this choice, including consent withdrawal, please see our Privacy Policy .

Reviewer #1: No

Reviewer #2: No

Reviewer #3: No

---

## [Author Response · Author response to Decision Letter 3]

1 Feb 2025

Abstract

• Line 42- its better to refer to factors associated with genital ulcers/discharge in the past 12m as opposed to STIs. I think one of the reviewers recommended a change in title but I think it should go beyond the title.

• Line 45- refer to prevalence of ulcers/discharge

We thank you for your comments, we have taken note and made changes throughout the manuscript.

Introduction

• Line 78- state whether the study of STI in the 8 health facilities included people who were symptomatic or asymptomatic.

We thank you for your comments. Prevalence of the stated infections were from patients who presented with genital symptoms. We have added this information as advised.

• Line 79- did the 2004 Tanzanian DHS test for STIs?

We thank you for your comments, the 2004 DHS did not test for STI, but asked survey respondents if they had STI in the last 12 months. We have edited the statement to reflect these were self-reported data.

• Line 82- add something about how STIs are managed in Tanzania. If its syndromic management, add information about its accuracy in predicting the presence of an STI among people with genital discharge or ulcers.

We thank you for your comments. We have added this information as advised.

Methods

• Line 114- might be more accurate to refer to the outcome as prevalence of ulcers/ discharge OR prevalence of STI-related symptoms. This is because the presence of an ulcer or discharge does not equate to an STI particularly for women or those with ulcers.

We thank you for your comments, we have changed our outcome variable to genital ulcers/discharge throughout the manuscript.

• Line 132 - the authors dont say what variables were used for weighting.

We thank you for your comments. Normally DHS uses cluster, household, individual , the variables related to selection probabilities at each stage of sampling as well as response rates at the household and individual level to calculate weights. These are then adjusted for non-response to create the final survey weights. This information has been added.

Results

• Line 175- prevalence of genital ulcer/discharge should be outcome and not STI prevalence. Please correct the results section

We thank you for your comments, we have changed the outcome to be genital ulcers/discharge throughout the manuscript.

• Line 199- 200- Add to the discussion why you found this? Generally STIs are more prevalenct in younger people but are more likely to be asymptomatic in that group. Prevalence of symptoms and not infection increases with age as people acquire immunity

We thank you for your comments, while this argument is plausible, literature indicates less STI among older individuals is more likely due to behavior and not necessarily immunity.

• Line 216 - women are more likely to have non-STI causes of ulcers/ discharge compared to males males eg vaginal thrush, bacterial vaginosis, allergies etc. Although women are also more likely to have prevalent STI infections as well, it will not to to the same extent as the likelihood of having ulcers/ discharge- Another systematic review showed steady burden of infections in SSA with the exception of chlamydia. (see Michalow, Julia, Hall, Lauren, Rowley, Jane, Anderson, Rebecca, Hayre, Quinton, Chico, R. Matthew, Edun, Olanrewaju, Knight, Jesse, Kuchukhidze, Salome, Majaya, Evidence, Reed, Domonique, Stevens, Oliver, Walters, Magdalene, Peters, Remco, Cori, Anne, Boily, Marie-Claude and Imai-Eaton, Jeffrey DOI: 10.1101/2024.12.16.24319070).

We thank you for your comments, we added this observations in the discussion

Discussion

• Lines 270- 277- increasing prevalence of symptoms does not imply increase in STI infection. This part of the discussion needs to be more nuanced. The two systematic reviews quoted show that the prevalence of specific STIs hasnt changed over time (except for chlamydia) among both symptomatics and aymptomatics. So unless there was testing in the surveys, there is no sufficient evidence of increase in infection but of increasing symptoms

We thank you for your comments, we have edited this section to reflects symptoms of STI and not necessarily STIs. DHS does not test people except for HIV. Other STI data collected is based on self-reported information on history of STI and associated symptoms.

• Line 278- but receiving an HIV result was not statistically significant in your adjusted model? Is there testing for STIs in Tanzania? A possible explanatiom for this may be that an HIV negative result is associated with continued high risk behaviour or doesnt change existing high risk behaviours hence the prevalence of STI-associated symptoms.

We thank you for your comments, Testing for STI in Tanzania is often done in high tier hospitals, majority of people are treated syndromically. Because our association between testing and receiving HIV results on genital ulcers/discharge was based on bivariate analysis, this section in the discussion has been deleted.

---

## [Editor Report · Decision Letter 3]

4 Feb 2025

Trend of genital ulcers and discharge and associated factors among survey respondents in Tanzania, 2004-2022: Analysis of demographic health surveys.

PONE-D-24-28121R3

Dear Dr. Ally,

We’re pleased to inform you that your manuscript has been judged scientifically suitable for publication and will be formally accepted for publication once it meets all outstanding technical requirements.

Kind regards,

Joel Msafiri Francis, MD, MS, PhD

Academic Editor

PLOS ONE
---

## [Editor Report · Acceptance letter]

PONE-D-24-28121R3

PLOS ONE

Dear Dr. Ally,

I'm pleased to inform you that your manuscript has been deemed suitable for publication in PLOS ONE. Congratulations! Your manuscript is now being handed over to our production team.

Kind regards,

on behalf of

Prof. Joel Msafiri Francis

Academic Editor

PLOS ONE